# Molecular Detection and Phylogeny of Tick-Borne Pathogens in Ticks Collected from Dogs in the Republic of Korea

**DOI:** 10.3390/pathogens10050613

**Published:** 2021-05-17

**Authors:** A-Tai Truong, Jinhyeong Noh, Yeojin Park, Hyun-Ji Seo, Keun-Ho Kim, Subin Min, Jiyeon Lim, Mi-Sun Yoo, Heung-Chul Kim, Terry A. Klein, Hyunkyoung Lee, Soon-Seek Yoon, Yun Sang Cho

**Affiliations:** 1Parasitic and Honeybee Disease Laboratory, Bacterial and Parasitic Disease Division, Animal and Plant Quarantine Agency, Gimcheon 39660, Korea; Taita@korea.kr (A.-T.T.); jhnoh@korea.kr (J.N.); pyj931027@naver.com (Y.P.); hyunj3589@korea.kr (H.-J.S.); northbear5@korea.kr (K.-H.K.); alstnqls5917@naver.com (S.M.); jiyeom75@hanmail.net (J.L.); msyoo99@korea.kr (M.-S.Y.); yoonss24@korea.kr (S.-S.Y.); 2Faculty of Biotechnology, Thai Nguyen University of Sciences, Thai Nguyen 250000, Vietnam; 3Force Health Protection and Prevention Medicine, Medical Department Activity-Korea, 65th Medical Brigade, Unit 15281, APO AP 96271-5281, USA; hungchol.kim2.ln@mail.mil (H.-C.K.); terry.a.klein2.civ@mail.mil (T.A.K.); 4Animal Pathodiagnostic Laboratory, Animal Disease Diagnostic Division, Animal and Plant Quarantine Agency, Gimcheon 39660, Korea; ieustina@korea.kr

**Keywords:** dog ticks, *Haemaphysalis longicornis*, *Ixodes nipponensis*, *Haemaphysalis* *flava*, anaplasmosis, Lyme borreliosis, Korea

## Abstract

Ticks are important vectors of various pathogens that result in clinical illnesses in humans and domestic and wild animals. Information regarding tick infestations and pathogens transmitted by ticks is important for the identification and prevention of disease. This study was a large-scale investigation of ticks collected from dogs and their associated environments in the Republic of Korea (ROK). It included detecting six prevalent tick-borne pathogens (*Anaplasma* spp., *A. platys*, *Borrelia* spp., *Babesia gibsoni*, *Ehrlichia canis*, and *E. chaffeensis*). A total of 2293 ticks (1110 pools) were collected. *Haemaphysalis longicornis* (98.60%) was the most frequently collected tick species*,* followed by *Ixodes nipponensis* (0.96%) and *H. flava* (0.44%). *Anaplasma* spp. (24/1110 tick pools; 2.16%) and *Borrelia* spp. (4/1110 tick pools; 0.36%) were detected. The phylogenetic analyses using 16S rRNA genes revealed that the *Anaplasma* spp. detected in this study were closely associated with *A. phagocytophilum* reported in humans and rodents in the ROK. *Borrelia* spp. showed phylogenetic relationships with *B. theileri* and *B. miyamotoi* in ticks and humans in Mali and Russia. These results demonstrate the importance of tick-borne disease surveillance and control in dogs in the ROK.

## 1. Introduction

Ticks are obligate blood-feeding parasites that transmit zoonotic tick-borne pathogens, including protozoa, viruses, and bacteria, to animal and human hosts [1,2]. Approximately 10% of known tick species are vectors of pathogens of medical and veterinary importance [3]. Some tick species are known vectors of one or several tick-borne diseases (TBDs), such as borreliosis (Lyme disease and *Borrelia* relapsing fever and other *Borrelia* spp. transmitted by *Ixodes* ticks), babesiosis (*Babesia* spp. transmitted by *Haemaphysalis* spp., *Rhipicephalus* spp., and *Dermacentor* spp.), ehrlichiosis (*Ehrlichia canis* genogroup transmitted by *Rhipicephalus* spp., *Amblyomma*
*americanum* [4,5,6], *Ixodes persulcatus*, *I. ovatus*, and *I. silvarum*), and anaplasmosis (*Anaplasma phagocytophilum* transmitted by *I. scapularis* and *I. pacificus* [7]). An understanding of the specific tick hosts and associated pathogens is important to identify the risks of TBDs for domestic animals and humans. 

Ticks are common parasites of domestic animals, including dogs, and have a high risk of transmitting tick-borne pathogens [8,9,10]. Dogs are reservoirs of some tick-borne pathogens [6]. They are the most common animal bred for various purposes, including pets and military dogs. Close contact with dogs may result in the transfer of ticks and TBDs to humans. Therefore, dogs may be considered sentinel animals for TBDs impacting human health [11,12,13,14]. Identifying the prevalence of dog ticks and associated pathogens provides an understanding of the distribution of tick-borne pathogens. It raises awareness of TBDs among pet owners and other people who contact dogs [15].

In the ROK, TBDs, such as Lyme disease, anaplasmosis, ehrlichiosis, tularemia, bartonellosis, and babesiosis, are of medical importance. In particular, the number of Lyme disease cases has rapidly increased since 2012 [16,17,18]. Molecular and serological detection methods have revealed that over 40% of dogs in the ROK are infected with pathogens that cause TBDs, including *A. phagocytophilum*, *E. canis*, *Borrelia burgdorferi*, *Babesia gibsoni*, *Dirofilaria immitis,* and *Mycoplasma haemocanis* [19,20]. The identification of reservoir hosts and potential vectors of the pathogenic agents is of interest. A previous study using molecular detection methods reported *I. nipponensis* ticks infected with *B. garinii* in dogs in the Gyeongsangbuk province, ROK [21]. However, the prevalence of ticks on dogs and tick-borne pathogens harbored by ticks collected from dogs in the ROK remains poorly investigated.

This study was part of a large-scale tick surveillance program of domestic pets, military working dogs, and stray dogs from shelter-associated environments in the ROK. Assays to detect six common tick-borne pathogens (*A. phagocytophilum*, *A. platys*, *Borrelia* spp., *Babesia gibsoni*, *E. canis*, and *E. chaffeensis*) were conducted. This study highlights the importance of the prevention of TBDs in dogs and humans in the ROK.

## 2. Results

### 2.1. Distribution of Dog Ticks in the ROK

A total of 2293 ticks categorized into 1110 tick pools were collected from 24 sites in 13 provinces or metropolitan cities in the ROK (Figure 1). Overall, 807 ticks (35.2%) were found on pet dogs, 624 (27.2%) on military dogs, 572 (24.95%) on stray dogs, and 290 (12.65%) in stray dog shelter environments (Table 1).

The highest number of ticks collected from dogs/dog shelters was observed in the northern region (1461 ticks, 63.72%), followed by the central region (767 ticks, 33.45%), and the southern region (65 ticks, 2.83%) (Table 2 and Figure 1).

### 2.2. Identification of Tick Species

The most commonly collected tick species was *H. longicornis* (98.60%; 2261/2293), followed by *I. nipponensis* (0.96%; 22/2293) and *H. flava* (0.44%; 10/2293) (Table 2 and Figure 2).

### 2.3. Detection of Tick-Borne Pathogens in Dog Ticks

*Anaplasma* spp. and *Borrelia* spp. were detected in the collected ticks using polymerase chain reaction (PCR). Detection of these pathogens was confirmed using 16S rRNA gene fragments of amplicons of 511 bp (*Anaplasma* spp.) and 714 bp (*Borrelia* spp.) (Figure 3).

Overall, 2.16% of the tick pools (24/1110 tick pools) contained *Anaplasma* spp. and 0.36% (4/1110 tick pools) contained *Borrelia* spp. *Anaplasma* spp. was detected in 22/1082 *H. longicornis* tick pools (2.03%) and 2/22 (9.09%) *I. nipponensis* tick pools (Table 3). *Anaplasma* spp. was detected in all stages (larvae, nymphs, and adults) of *H. longicornis*, but only in adults of *I. nipponensis*. *Borrelia* spp. was detected in 3/22 (13.64%) adult *I. nipponensis* tick pools and 1/1082 (0.09%) adult *H. longicornis* tick pools (Table 3).

Half of the tick pools found to be positive for *Anaplasma* spp. (12/24 tick pools) were collected from the northern region of the ROK, including 11 tick pools collected from military dogs and one tick pool collected from a pet. In the central region, eight tick pools were positive for *Anaplasma* spp., including five collected from stray dogs and three collected from pets. All four tick pools from the southern region that were positive for *Anaplasma* spp. were collected from stray dogs (Table 4).

*Borrelia* spp. was only detected in ticks collected in the northern and central regions and was predominantly detected in ticks collected from pet dogs (75%, 3/4). However, one positive pool (25%, 1/4) was collected from a stray dog (Table 4).

### 2.4. Sequencing and Phylogenetic Analysis of Tick-Borne Pathogens

*Anaplasma* spp. detection was confirmed using sequencing analyses. The sequence of *Anaplasma* spp. detected in each of the 24 positive pools was 98.01–100% identical to previously deposited sequences of *A. phagocytophilum* in the National Center for Biotechnology Information (NCBI) GenBank database. In addition, phylogenetic analyses revealed a close relationship between the *Anaplasma* spp. detected in this study with previously reported *A. phagocytophilum* found in rodents, raccoon dogs, domestic/stray/military working dogs, and humans in the ROK, USA, Poland, Slovenia, and Norway (Figure 4).

The sequences of the *Borrelia* spp. detected in four tick pools in this study were 98.62–100% identical to previously reported sequences of *Borrelia* spp. listed in the NCBI database. Phylogenetic analysis revealed that the *Borrelia* strain detected in tick pool 18D249 demonstrated a close relationship with *B. theileri* previously reported in ticks and cattle from Mali and Egypt, respectively. The *Borrelia* strains detected in the 18D12 and 18C04 tick pools were closely related to *B. miyamotoi* (*Borrelia* relapsing fever) detected in a human in Russia. The 18C01 tick pool strain was similar to *B. garinii*, *B. tanuki*, and *B. bissettii* strains from Russia, Japan, and the USA, respectively (Figure 5).

## 3. Discussion

This study determined the distribution of tick-borne pathogens detected in ticks collected from pet, stray, and military working dogs and dog shelter environments in the ROK. Three tick species (*H. longicornis, I. nipponensis*, and *H. flava*) were identified. *H. longicornis*, which is commonly associated with grass/herbaceous vegetation habitats, was the most commonly collected species. *H. flava*, which is commonly found in forested habitats, was notably less prevalent. *I**. nipponensis*, which is associated with both grass/herbaceous vegetation and forested habitats, was also less prevalent [22]. As dogs are more likely to enter grass/herbaceous vegetation than forests, they are exposed to *H. longicornis* ticks more frequently in the ROK [22,23,24]. However, a previous study reported that only *I. nipponensis* ticks were collected from dogs in the Gyeongsangbuk province [21].

While *A. phagocytophilum*, *Borrelia* spp., *Ehrlichia* spp., and *Babesia* spp. are present in the ROK, only *Anaplasma* spp. and *Borrelia* spp. were detected in this study. Few cases of *A. phagocytophilum* in humans in the ROK have been reported [25]. The composition of tick-borne pathogens in dog ticks varies worldwide. *Rickettsia* spp., *Borrelia* spp., *A. phagocytophilum*, and *Babesia* sp. have been reported in Latvia [15]. *A. phagocytophilum*, *Ehrlichia canis*, and *Babesia gibsoni* have been reported in Taiwan [26]. *E. canis*, *Hepatozoon canis*, *Rickettsia* spp., *Candidatus* Neoehrlichia mikurensis and *A. platys* have been reported in Nigeria [27]. Five genera of pathogens (*Anaplasma* spp., *Babesia* spp., *Borrelia* spp., *Ehrlichia* spp., and *Theileria cervi*) have been reported in dog ticks in Russia [28]. The pathogens detected in this study were consistent with previously reported TBDs in dogs [19,20]. These results are useful for the surveillance of TBDs present in dogs that may impact the transmission of these pathogens to dog owners or handlers throughout the ROK.

*A. phagocytophilum* has been found in various tick species worldwide [29,30,31]. In this study, only *H. longicornis* and *I. nipponensis* ticks were positive for *Anaplasma* spp., with a predominance in *H. longicornis* ticks, which may be due to the lower numbers of *I. nipponensis* and *H. flava* that were collected in this study. These findings are consistent with a previous study conducted in the ROK [32]. *H. longicornis* ticks carrying *Anaplasma* spp. were collected from pets, military working dogs, and stray dogs, but not from vegetation surrounding dog shelters. The phylogenic analyses demonstrated a close relationship between the *Anaplasma* spp. detected in this study and previously reported *A. phagocytophilum* strains from dogs and humans in the ROK. Therefore, the transmission of *A. phagocytophilum* to humans may result from exposure to ticks on pets. Therefore, pet owners, dog shelter workers, and handlers of military working dogs should be educated regarding the potential of transmitting anaplasmosis.

The primary vectors of *Borrelia* spp. are *Ixodes* spp. [33,34]. In this study, *H. longicornis* ticks were also found to be vectors of *Borrelia* spp. *H. longicornis* ticks in this study may have fed on a *Borrelia*-positive animal, resulting in the detection of the pathogen. However, whether *H. longicornis* ticks are a vector of *Borrelia* spp. has not been determined. The sequence analyses demonstrated that there are at least three species or strains of *Borrelia* spp. in the ROK, including *B. theileri*, *B. miyamotoi*, and an unidentified *Borrelia* sp. Additional analyses using other genes (such as the flagellin gene and PCR-restriction fragment length polymorphism) are necessary to determine the specific phylogenetic identification of *Borrelia* spp. [35].

In this study, nationwide surveillance of dog ticks and tick-borne pathogens was conducted, and three tick species were collected. *H.*
*l**ongicornis* was the most prevalent tick species detected in this study, followed by *I. nipponensis* and *H. flava*. *Anaplasma* spp. and *Borrelia* spp. were detected on *H. longicornis* and *I. nipponensis* ticks only. Phylogenetic analyses suggested that at least two species of *Borrelia (B. theileri* and *B. miyamotoi)* were present. In contrast, a third species of *Borrelia* detected in this study remains unidentified. This study demonstrates that dogs and dog owners/handlers in the ROK have a relatively high risk of becoming infected with *Anaplasma* spp. or *Borrelia* spp. Therefore, disease screening is important not only to determine the distribution and prevalence of dog TBDs but also to understand the potential impact on veterinary and human health.

## 4. Materials and Methods

### 4.1. Collection of Ticks

Ticks were collected from pet dogs, military working dogs, stray dogs, and vegetation surrounding dog shelters in 13 provinces and metropolitan cities in the ROK from 2017 to 2018 (Figure 1). Ticks were removed using fine forceps to secure the tick mouthparts at the point of attachment and gently pulling the tick out to avoid breaking off the mouthparts. After removal, the ticks were transferred to a 50 mL conical tube and placed in a cooler to be transferred to the Parasitic and Honeybee Disease Laboratory, Animal and Plant Quarantine Agency for species identification and pathogen detection.

### 4.2. Identification of Tick Species

Ticks were identified using morphological keys [36,37,38] then placed in 1.5 mL cryovials according to species, host, date, and stage of development. The samples were preserved in 70% ethanol and stored at −80 °C until they were used to detect tick-borne pathogens. The tick pools each included 1–5 adult ticks, 1–30 nymphs, or 1–50 larvae.

### 4.3. Isolation of Tick Nucleic Acids

Total nucleic acid extraction was performed with the Maxwell RSC viral total nucleic acid purification kit (Promega, Madison, WI, USA) for each tick pool. Briefly, 330 µL of lysis buffer and six stainless steel beads with diameters of 2.381 mm (SNC, Hanam, Korea) were used to homogenize the ticks with a Precellys 24 tissue homogenizer (Bertin Instruments, Montigny-le-Bretonux, France). The tick homogenate was placed in a Maxwell RSC instrument (Promega, Madison, WI, USA) according to the manufacturer’s instructions. The purification of the total nucleic acids was conducted automatically. Finally, 50 µL of total nucleic acids were acquired from each pool and used to detect tick-borne pathogens.

### 4.4. Detection of Tick-Borne Pathogens

Conventional PCR was performed to detect six tick-borne pathogens: *Anaplasma* spp., *A. platys*, *E. canis*, *E. chaffeensis*, *Borrelia* spp., and *Babesia gibsoni*. Specific primers of each target agent (Table 5) and the AccuPower ProFi Taq PCR PreMix (Bioneer, Daejeon, Korea) were utilized. Each 20 µL reaction mix included 1 µL (10 pmol) of each primer, 5 µL of total nucleic acids, and 13 µL of double-distilled water (ddH_2_O). The PCR conditions for the detection of each pathogen are shown in Table 5.

### 4.5. Sequencing and Phylogenetic Analysis

The products of conventional PCR were analyzed using agarose gel electrophoresis. After electrophoresis, the PCR products were purified using a QIA quick purification kit (Qiagen, Hilden, Germany) and sequenced by Macrogen (Seoul, Korea). The sequences of the *Anaplasma* spp. and *Borrelia* spp. detected in this study were deposited in the NCBI database with accession numbers of MW793414-MW793437 (*Anaplasma* spp.) and MW793441-MW793444 (*Borrelia* spp.). The generated sequences were compared to previously reported sequences in the NCBI GenBank database. Identical sequences of the *Anaplasma* spp. and *Borrelia* spp. were aligned using Clustal X version 2.0 [45]. Maximum-likelihood phylogenetic trees were created using the Kimura 2-parameter model, gamma distribution, and bootstrapping 1000 times with MEGA7 software [46].

## Figures and Tables

**Figure 1 pathogens-10-00613-f001:**
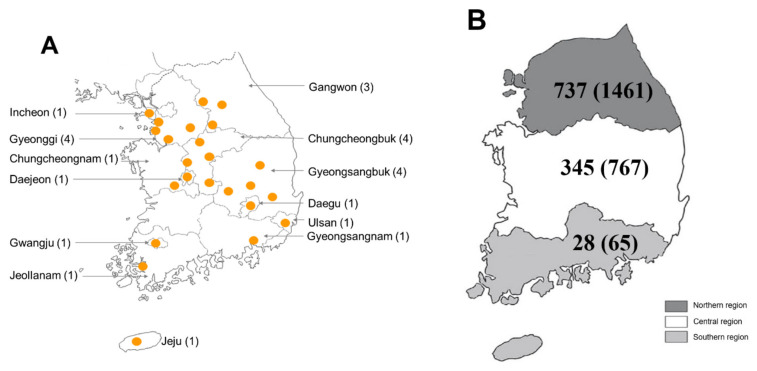
Collection of ticks from dogs and vegetation at dog shelters in the ROK. Ticks were collected at 24 sites in 13 provinces or metropolitan areas. The number of sites per province or metropolitan area is shown in parentheses (**A**). The numbers of tick pools collected from the northern, central, and southern regions are shown, with the number of collected ticks shown in parentheses (**B**).

**Figure 2 pathogens-10-00613-f002:**
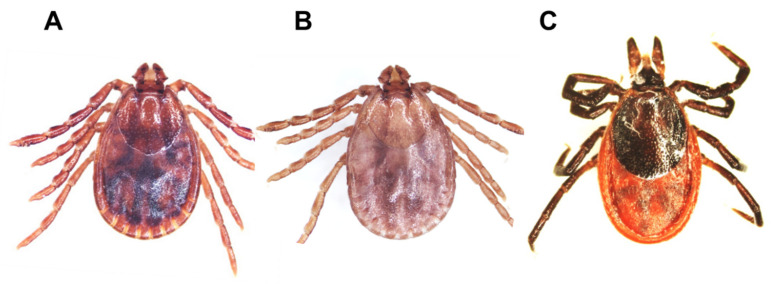
Morphological identification of three tick species collected from dogs in the ROK. *Haemaphysalis longicornis* female (**A**), *H. flava* female (**B**), and *Ixodes nipponensis* female (**C**) are shown.

**Figure 3 pathogens-10-00613-f003:**
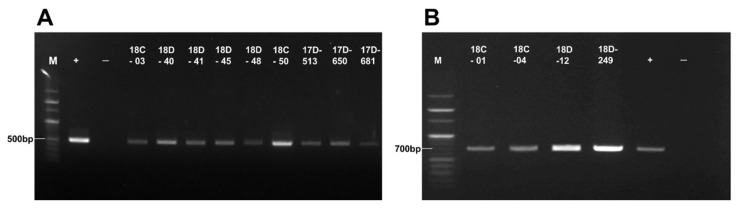
Detection of *Anaplasma* spp. and *Borrelia* spp. from dog ticks collected in the ROK. *Anaplasma* spp. were detected by the amplification of a 16S rRNA gene with a band of 511 bp. Nine of 24 tick pools positive for *Anaplasma* spp. are shown (**A**). Amplification of the *Borrelia* spp. 16S rRNA gene (714 bp) was observed in four tick pools (**B**). A positive control using recombinant DNA (+) and negative control (-) using no DNA template are shown. M represents the 100 bp DNA marker.

**Figure 4 pathogens-10-00613-f004:**
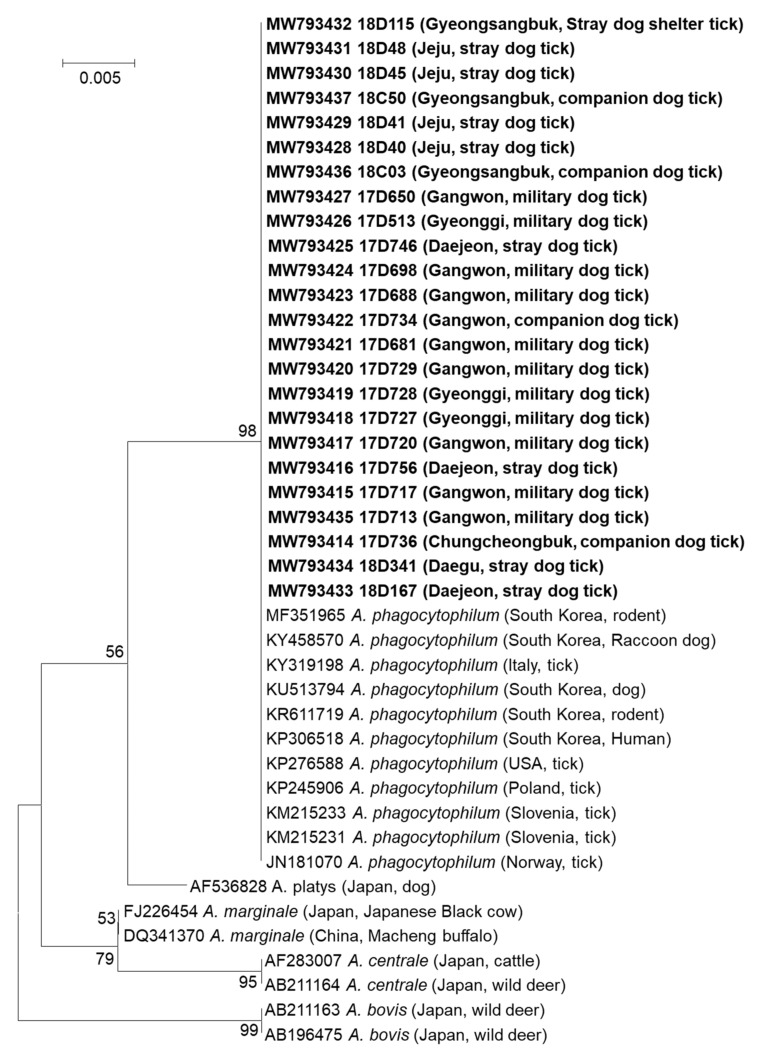
The phylogenetic analysis of *Anaplasma* spp. detected in ticks collected from dogs in the ROK. The maximum-likelihood tree was created based on the nucleotide sequences of 16S RNA (511 bp) of *Anaplasma* spp. detected in this study and other countries. MEGA7 software with 1000 bootstrap replications was used to create the phylogenetic tree. The NCBI accession numbers and names of the *Anaplasma* spp. positive pools are shown in bold. The names of the province or metropolitan city and host of the detected *Anaplasma* spp. are shown in parentheses. Reference strains of *Anaplasma* spp. with NCBI accession numbers and country of detection and host are also shown.

**Figure 5 pathogens-10-00613-f005:**
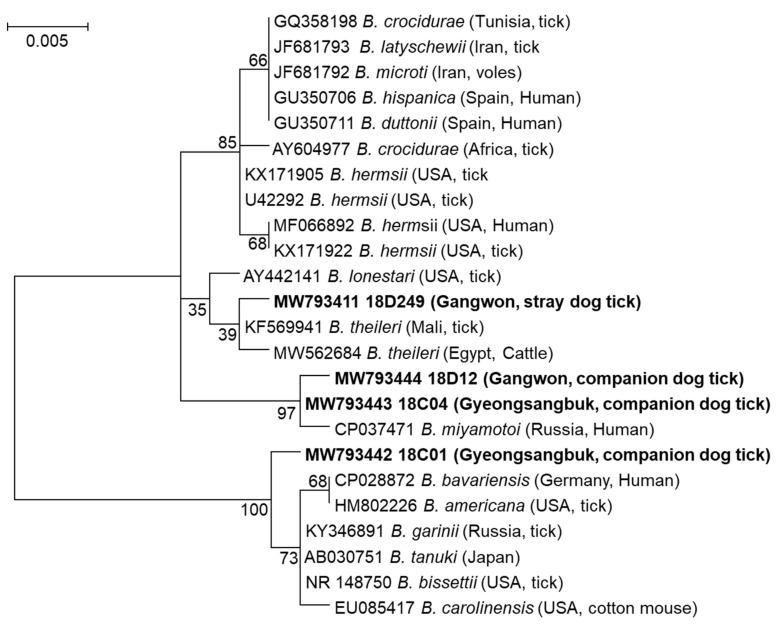
Phylogenetic analysis of *Borrelia* spp. detected in ticks collected from dogs in the ROK. The maximum-likelihood tree presenting the phylogenetic relationship between detected *Borrelia* spp. and previously reported strains was generated based on the nucleotide sequences of 16S rRNA (714 bp) of *Borrelia* spp. detected in this study. MEGA7 software with 1000 bootstrap replications was used. The NCBI accession numbers and names of positive *Borrelia* spp. tick pools are in bold. The country, province, or metropolitan city and host of the detected *Borrelia* spp. are shown in parentheses. Reference strains of *Borrelia* spp. with NCBI accession numbers and country of detection and host are also shown.

**Table 1 pathogens-10-00613-t001:** Identification of dog ticks in the ROK.

Species	Stage	Tick Pool (Number of Ticks)				Total
		Pet Dogs	Stray Dogs	Dog Shelters ^1^	Military Working Dogs	
*Haemaphysalis longicornis*	Larvae	4 (49)	26 (126)	4 (166)	0	34 (341)
	Nymph	73 (606)	51 (149)	9 (93)	15 (110)	148 (958)
	Adult	111 (129)	282 (295)	29 (29)	478 (509)	900 (962)
	***Sub total***	***188 (784)***	***359 (570)***	***42 (288)***	***493 (619)***	***1082 (2261)***
*H. flava*	Larvae	0	0	0	0	0
	Nymph	1 (3)	0	1 (1)	1 (3)	3 (7)
	Adult	2 (2)	0	0	1 (1)	3 (3)
	***Sub total***	***3 (5)***	***0 (0)***	***1 (1)***	***2 (4)***	***6 (10)***
*Ixodes nipponensis*	Larvae	0	0	0	0	0
	Nymph	0	0	0	0	0
	Adult	18 (18)	2 (2)	1 (1)	1 (1)	22 (22)
	***Sub total***	***18 (18)***	***2 (2)***	***1 (1)***	***1 (1)***	***22 (22)***
**Total**		**209 (807)**	**361 (572)**	**44 (290)**	**496 (624)**	**1110 (2293)**

^1^ Ticks were collected from vegetation bordering stray dog shelter pens. Data are presented as numbers.

**Table 2 pathogens-10-00613-t002:** Numbers of ticks collected from dogs and dog shelters by latitudinal region.

Region	Site	Species			Total (%)
		*Haemaphysalis longicornis*	*H. flava*	*Ixodes nipponensis*	
**Northern**	Pet dogs	552	4	12	568
	Stray dogs	189	0	1	190
	Dog shelters ^1^	100	0	0	100
	Military working dogs	599	3	1	603
	***Subtotal***	***1440 (62.80)***	***7 (0.31)***	***14 (0.61)***	***1461 (63.72)***
**Central**	Pet dogs	228	1	6	235
	Stray dogs	321	0	1	322
	Dog shelters	188	1	1	190
	Military working dogs	20	0	0	20
	***Subtotal***	***757 (33.01)***	***2 (0.09)***	***8 (0.35)***	***767 (33.45)***
**Southern**	Pet dogs	4	0	0	4
	Stray dogs	60	0	0	60
	Dog shelters	0	0	0	0
	Military working dogs	0	1	0	1
	***Subtotal***	***64 (2.79)***	***1 (0.04)***	***0***	***65 (2.83)***
**Total (%)**		**2261 (98.60)**	**10 (0.44)**	**22 (0.96)**	**2293 (100)**

^1^ Ticks were collected from vegetation bordering stray dog shelter pens. Data are presented as numbers (percentage).

**Table 3 pathogens-10-00613-t003:** Pathogens detected by molecular (PCR).

Scheme	Stage	Number of Tick Pools	Species					
			*Anaplasma* spp.	*A. platys*	*E. canis*	*E. chaffeensis*	*Borrelia* spp.	*B. gibsoni*
*Haemaphysalis longicornis*	Larvae	34	3	-	-	-	-	-
	Nymph	148	5	-	-	-	-	-
	Adult	900	14	-	-	-	1	-
	***Subtotal***	***1082***	***22 (2.03)***	-	-	-	***1 (0.09)***	-
*H. flava*	Larvae	0	-	-	-	-	-	-
	Nymph	3	-	-	-	-	-	-
	Adult	3	-	-	-	-	-	-
	***Subtotal***	***6***	-	-	-	-	-	-
*Ixodes nipponensis*	Larvae	0	-	-	-	-	-	-
	Nymph	0	-	-	-	-	-	-
	Adult	22	2	-	-	-	3	-
	***Subtotal***	***22***	***2 (9.09)***	-	-	-	***3 (13.64)***	-
**Total**		**1110**	**24 (2.16)**	**0**	**0**	**0**	**4 (0.36)**	**0**

Data are presented as numbers (percentage).

**Table 4 pathogens-10-00613-t004:** Distribution of tick-borne pathogens in dog ticks collected in the ROK.

Region	Site	Number of Tick Pools	Species					
			*Anaplasma* spp.	*A. platys*	*E. canis*	*E. chaffeensis*	*Borrelia* spp.	*B. gibsoni*
**Northern**	Pet dogs	84	1 (1.19)	-	-	-	1 (1.19)	-
	Stray dogs	169	-	-	-	-	1 (0.59)	-
	Dog shelters ^1^	2	-	-	-	-	-	-
	Military working dogs	482	11 (2.28)	-	-	-	-	-
	***Subtotal***	***737***	***12 (1.63)***	-	-	-	***2 (0.27)***	-
**Central**	Pet dogs	121	3 (2.48)	-	-	-	2 (1.65)	-
	Stray dogs	169	5 (2.96)	-	-	-	-	-
	Dog shelters	42	-	-	-	-	-	-
	Military working dogs	13	-	-	-	-	-	-
	***Subtotal***	***345***	***8 (2.32)***	-	-	-	***2 (0.58)***	-
**Southern**	Pet dogs	4	-	-	-	-	-	-
	Stray dogs	23	4 (17.39)	-	-	-	-	-
	Dog shelters	0	-	-	-	-	-	-
	Military working dogs	1	-	-	-	-	-	-
	***Subtotal***	***28***	***4 (14.29)***	-	-	-	***0***	-
**Total (%)**		**1110**	**24 (2.16)**	**0**	**0**	**0**	**4 (0.36)**	**0**

^1^ Ticks were collected from vegetation bordering stray dog shelter pens. Data are presented as numbers (percentage).

**Table 5 pathogens-10-00613-t005:** Primers used for the detection of tick-borne pathogens.

Pathogens	Primers	Sequences (5′-3′)	Target Gene (bp)	PCR Conditions	References
*Anaplasma* spp.	PITA-F	GTCGAACGGATTATTCTTTA	16S rRNA (511)	95 °C (5 min); 37 cycles of 95 °C (30 s), 50 °C (30 s), and 72 °C (40 s); 72 °C (7 min)	[39]
	PITA-R	TTCACCTTTAACTTACCGAA			
*A. platys*	EPLAT5	TTTGTCGTAGCTTGCTATGAT	16S rRNA (359)	95 °C (5 min); 37 cycles of 95 °C (30 s), 53 °C (30 s), and 72 °C (30 s); 72 °C (7 min)	[40]
	EPLAT3	CTTCTGTGGGTACCGTC			
*Ehrlichia canis*	ECAN5	GCAAATTATTTATAGCCTCTGGCTATAG	16S rRNA (365)	95 °C (5 min); 37 cycles of 95 °C (30 s), 56 °C (30 s), and 72 °C (30 s); 72 °C (7 min)	[41,42]
	HE3	TTATAGGTACCGTCATTATCTTCCCTA			
*E. chaffeensis*	HE1	ACAATATTGCTTATAACCTTTTGGTTATA	16S rRNA (390)	95 °C (5 min); 37 cycles of 95 °C (30 s), 56 °C (30 s), and 72 °C (30 s); 72 °C (7 min)	
	HE3	TTATAGGTACCGTCATTATCTTCCCTA			
*Borrelia* spp.	B3	GCAGCTAAGAATCTTCCGCA	16S rRNA (714)	95 °C (5 min); 37 cycles of 95 °C (30 s), 58 °C (30 s), and 72 °C (1 min); 72 °C (7 min)	[43]
	B6	CAACCATGCAGCACCTGTATAT			
*Babesia gibsoni*	PIRO-F	AGTCATATGCTTGTCTTA	18S rRNA (500)	95 °C (5 min); 37 cycles of 95 °C (30 s), 47 °C (30 s), and 72 °C (40 s); 72 °C (7 min)	[44]
	PIRO-R	CCATCATTCCAATTACAA			

## Data Availability

The data presented in this study are available in the manuscript.

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
