# Peer review of "Molecular Detection and Phylogeny of Tick-Borne Pathogens in Ticks Collected from Dogs in the Republic of Korea"

_pathogens, 2021, doi:10.3390/pathogens10050613_

Round 1
Reviewer 1 Report
The authors collected ticks from dogs and surroundings across the Republic of Korea. They identified 2293 ticks as Ixodes nipponensis (0.93%), Haemaphysalis flava (0.44%) and Haemaphysalis longicornis (98.6%), the latter being the most prevalent. They also tested these ticks for tick-borne pathogens using PCR and sequencing by sampling 1110 tick pools. They detected Anaplasma phagocytophilum (2.16% tick pools positive), Borrelia spp. (0.36% tick pools positive). The pathogens were closely related to other sequences in the databank. They conclude that surveillance of dogs and their ticks as sentinels for human diseases may be a useful strategy. Overall the study reads very well and will be of interest to researchers in this field. The sampling, identification and molecular analysis seems to be equivalent to standard practices in this field. I have very little to add to improve the manuscript. I congratulate the authors on a very well described study. Some minor comments follow below.
Table 3: There seems to be an interesting gradation of ticks from the north to the south of the country with more ticks found in the north and the least in the south. The authors did not comment on this. Also, the majority of ticks in the south were found on stray dogs, while in the north ticks are found mostly on pet and military dogs. Is there a reason for this peculiar geographic distribution of ticks and their host predilection?
Page 3, Line 85-87: Species names should be italicized. Please check whole document.
Author Response
Reviewer 1: “Table 3: There seems to be an interesting gradation of ticks from the north to the south of the country with more ticks found in the north and the least in the south. The authors did not comment on this.”
Authors’ Reply: Thank you for your comments. The number of ticks shown in this study related to the number of sample collection sites and number of dogs were inspected. A larger number of dogs were assessed in Northern and Central regions therefore more ticks were collected compared to that than in the Southern region. The data of ticks collected in this study might be not sufficient to draw a conclusion about the regional distribution of ticks in Korea. Therefore, it was not commented in the study.
Reviewer 1: “Table 3: The majority of ticks in the south were found on stray dogs, while in the north ticks are found mostly on pet and military dogs. Is there a reason for this peculiar geographic distribution of ticks and their host predilection?”
Authors’ Reply: Thank you for your question. The sample collection in this study was done in collaboration with companion animal welfare centers in Korea. Collection of ticks from stray dogs and dog shelters were conducted in all three regions from the rescued dogs. However, in the case of pet dogs, a limited number of animal protection centers in the Southern region participated in this project therefore the number of ticks collected from pet dogs is less than in the Northern and Central regions. For the tick collection from military dogs, the military campuses with dog training for military purposes were all almost located in Northern region in Korea. Therefore, the ticks collected from military dogs were majorly from the Northern region.
Reviewer 1: “Page 3, Line 85-87: Species names should be italicized. Please check whole document.”
Authors’ Reply: Thank you for your suggestion. The species name of tick and pathogens in whole manuscript were checked and italicized.
Reviewer 2 Report
The manuscript is very well written. The results are relevant and interesting. English sounds great, even though it is not my native language.
My only request is that the 24 positive samples for A. phagocytophilum be retested, in order to sequence the complete 16S gene. My experience with 16S sequences from members of the Anaplasmataceae family says that sequences smaller than 700 nucleotides are not able to distinguish species, since this gene is very conserved.
As many positive ticks have not been detected, it is extremely important that this confirmation is carried out.
Author Response
Reviewer 2: “My only request is that the 24 positive samples for A. phagocytophilum be retested, in order to sequence the complete 16S gene. My experience with 16S sequences from members of the Anaplasmataceae family says that sequences smaller than 700 nucleotides are not able to distinguish species, since this gene is very conserved. As many positive ticks have not been detected, it is extremely important that this confirmation is carried out.”
Authors’ Reply: Thank you for your suggestion. The complete sequence of 16S gene could be important to identify more detail about the detected A. phagocytophilum. Unfortunately, the extracted DNAs of the 24 positive samples were completely used for the study. Therefore, additional experiment for amplification of complete 16S gene was not able to be conducted. However, result of comparison between the amplified 16S sequences and those from NCBI showed the highest identity (98.01-100%) with A. phagocytophilum. Therefore, we believe that the result of A. phagocytophilum detection showing in this study is accurate.
Reviewer 3 Report
The major limitation of the present study is testing engorged ticks. In the absence of host blood testing, the authors can’t determinate if the DNA was originating from tick or host’s blood.
Minor Comments:
- Line 41: Granulocytic anaplasmosis, vector? Add reference.
- Line 61: Maybe poorly investigated not unknown.
- Line 85: The first sentence belongs to methods.
- Lines 92-95: Belongs to methods.
Author Response
Reviewer 3: “The major limitation of the present study is testing engorged ticks. In the absence of host blood testing the authors can’t determinate if the DNA was originating from tick or host’s blood.”
Authors’ Reply: Thank you for your comments. After species identification, the tick samples were preserved in 70% ethanol and stored at -80℃ until they were used for DNA isolation; the additional information was added in the manuscript line 222-223. During the preservation of sample in ethanol the host blood attached (if have) on the ticks could be dissolved in the ethanol solution. Therefore, the probability of DNA originated from host’s blood and affect the detected result could be very low.
Round 2
Reviewer 2 Report
I cannot agree with the attestation that the 16S sequences originating in the present work are considered to be from A. phagocytophilum. In addition, the authors claim that the sequences range from 98 to 100%. This means considering in my experience that it may not be A. phagocytophilum because 98% is a very large variation for the 16S gene and may have originated from another species of Anaplasma.
I suggest, however, that the sequences be named as Anaplasma spp.
Author Response
Reviewer 2: “My only request is that the 24 positive samples for A. phagocytophilum be retested, in order to sequence the complete 16S gene. My experience with 16S sequences from members of the Anaplasmataceae family says that sequences smaller than 700 nucleotides are not able to distinguish species, since this gene is very conserved. As many positive ticks have not been detected, it is extremely important that this confirmation is carried out.”
Authors’ Reply: Thank you for your suggestion. The complete sequence of 16S gene could be important to identify the detected A. phagocytophilum in this study. Unfortunately, the extracted DNAs of the 24 positive samples were completely used for the study. Additional experiment for amplification of complete 16S gene was not able to be conducted. Therefore, results of Anaplasma detection in this study were changed to Anaplasma spp. in the entire manuscript.
L19, 22, 24, 93, 95, 97, 98, 100, 112, 115, 117, 126, 127, 130, 172, 186, 189, 191, 208, 213, 242, 253, 254, 257, Table 3, Table 4, and Table 5:
‘Anaplasma phagocytophilum’ was changed to ‘Anaplasma spp.’.